# Cerebral Microdialysis in Aneurismal Subarachnoid Hemorrhage Patient Reveals a Detrimental Shift in Brain Energy Metabolism, Despite Normal Perfusion Pressure

**DOI:** 10.3390/metabo10090341

**Published:** 2020-08-24

**Authors:** Frederik Nielsen, Pernille Haure, Jacob Madsen, Birgitte Steenfeldt Nielsen, Carsten Reides Bjarkam

**Affiliations:** 1Department of Neurosurgery, Aalborg University Hospital, 9000 Aalborg, Denmark; c.bjarkam@rn.dk; 2NeuroIntensive Care Unit, Aalborg University Hospital, 9000 Aalborg, Denmark; p.haure@rn.dk (P.H.); jama@rn.dk (J.M.); birgitte.nielsen@rn.dk (B.S.N.)

**Keywords:** mitochondrial dysfunction, SAH, CPP

## Abstract

The present case study concerns a patient admitted to our neuro-intensive care unit with a severe aneurismal subarachnoid hemorrhage rebleeding. The patient was equipped with multimodal neuromonitoring, including cerebral microdialysis. During the neuro-intensive care unit, there was a gradual decrease in cerebral perfusion pressure, which was within normally accepted levels, correlated to a detrimental shift in cerebral metabolism, from mitochondrial dysfunction to an ischemic pattern. Subsequently, the clinical and paraclinical status of the patient worsened. The present case highlights how the dynamic assessment of cerebral metabolic patterns and the concept of mitochondrial dysfunction can be relevant in the day-to-day clinical setting, to evaluate and optimize basic, well-known physiological parameters, such as cerebral perfusion pressure.

## Patient Case: Cerebral Microdialysis in Aneurismal Subarachnoid Hemorrhage Patient Reveals a Detrimental Shift in Brain Energy Metabolism, Despite Normal Perfusion Pressure

Aneurismal subarachnoid hemorrhage (aSAH) compromises cerebrovascular compliance and autoregulation, whereby cerebral blood flow, and in consequence, the cerebral supply of oxygen and glucose, becomes largely dependent on adequate cerebral perfusion pressure (CPP), through the regulation of mean arterial blood pressure (MAP) and intracranial pressure (ICP) [1].

Consequently, insufficient CPP among aSAH patients has been linked to an increased risk for thrombosis, metabolic dysfunction, delayed cerebral ischemia (DCI) and poor clinical outcome [2,3]. Accordingly, current clinical guidelines generally advise keeping CPP above 70 mmHg for aSAH patients, and adjusting it based on clinical and paraclinical information [4,5,6].

The present case story illustrates how cerebral microdialysis (CMD) enabled the recognition of a detrimental shift in cerebral metabolism and subsequent clinical worsening in an aSAH patient, despite CPP values kept above 69 mmHg.

A middle-aged woman was referred after she had experienced a sudden debut of central headache and episodic blurring of the vision 3 days previously. A CT-scan of the cerebrum including angiography revealed fresh blood in the anterior interhemispheric subarachnoidal space (Fisher grade I), due to a broad-based aneurism of the anterior communicating artery (Figure 1A,B). Her initial Glasgow coma scale (GCS) was normal, at 15. Blood pressure was stabilized and oral Nimodipine was administered prophylactically. She was treated with the surgical clipping of the aneurism (Figure 1C). The control CT-cerebrum was without any ischemic areas, and she was admitted under stable conditions to the neurosurgery ward for observation. At day 4 after surgery, she developed CT-verified hydrocephalus, which was successfully treated by the placement of an external ventricular drain (EVD).

Five days after admission and surgical clipping, the patient developed a sudden loss of consciousness (GCS 3), due to a CT verified aneurismal rebleeding (Figure 1D). Patient consent was obtained for this case report. The patient was intubated, and an angiography showed a remodeling of the aneurism that allowed immediate endovascular coiling. Subsequent to coiling, an ICP-probe (Camino, Natus, Middleton, WI, USA) was placed. It showed increased intracranial pressure at 20–30 mmHg, and consequently, surgical evacuation of the intraventricular hematoma was performed to address the acute risk of herniation (Figure 1E).

The patient was transferred to the neuro-ICU, where she was equipped with a galvanometer (Licox, Integra, Upper Saddle River, NJ, USA) for measuring cerebral oxygen tension, and a cerebral microdialysis (CMD) catheter (Iscus flex, Mdialysis, Stockholm, Sweden) for measuring brain glucose metabolism (glucose, pyruvate and lactate), cerebral excitotoxicity (glutamate) and local cerebral cell damage (glycerol).

All probes were placed in the anterior watershed area between the frontal and middle cerebral artery on the non-dominant right side. Control-CT angiography showed a correctly placed coil and no signs of vasospasms. The brain tissue around the probes appeared radiological intact (Figure 1F).

During the first days after the rebleeding (day 7–13 after admission), her clinical condition was stable, but without improvement. The time period was characterized by CMD values with a lactate-pyruvate ratio (L/P-ratio), ranging from 25 to 35 combined with CMD-pyruvate between 80 and 160 µmol/L (Figure 2B). CMD-glucose ranged from 1 to 1.5 mmol/L, combined with blood glucose stabilized around 8 mmol/L (Figure 1C).

This corresponds to the pattern of mitochondrial dysfunction, which is defined as an elevated L/P-ratio combined with normal or increased pyruvate. It is caused by dysfunctional mitochondria, leading to increased anerobic lactate formation, in order to maintain glycolysis. The condition is assumed to exist on a continuum with the ischemic metabolic pattern [7,8].

A CT angiography on day 9 after admission indicated possible vasospasms, which was treated by intra-arterial administration of nimodipine, but with little effect on the clinical condition.

At day 12 after admission, a CT-angiography showed no detectable vasospasms, the clinical status was stable, ICP was below 10 mmHg (Figure 2A) and PbtO2 values were above 30 mmHg (Figure 2E). Therefore, the MAP was gradually decreased from 110 mmHg under close supervision over the following days (Figure 2A).

At day 15 after admission, the doctor on duty noted a sharply rising L/P-ratio with an apex value of 52.8, a CMD-glucose of 0.67 mmol/L, but stable PbtO2 at 26 mmHg and ICP at 8 mmHg. It was furthermore noted that the CPP had been gradually lowered over the preceding 3 days, from initially 110 mmHg to 69 mmHg. Parallel to the gradual decrease in CPP, a trend of decreasing CMD-glucose and CMD-pyruvate were identified despite a slow increase in blood glucose concentrations. The difference in both pyruvate and L/P ratio was statistically significant (*p* < 0.05), when comparing 60 CMD samples (corresponding to 2.5 days in each direction), before and after the L/P-ratio value of 52.8 (Figure 2A–C).

Subsequently, the CPP of the patient was increased, and the L/P-ratio dropped to 35–40. Despite this, CMD-pyruvate remained primarily below 80 µmol/L, and it was necessary to increase blood glucose, to maintain CMD-glucose above 1 mmol/L (Figure 2A–C). Blood glucose also became increasingly difficult to keep at a stable level below 10 mmol/L. Further, PbtO2 dropped from values around 30–35 mmHg to 20–25 mmHg, despite an increase in PaO2 and CPP (Figure 2E) and rigorous ventilator optimization. A CT scan and CT-angiography showed no vasospasms, but a growing hypodense area in the right hemisphere (Figure 1G), and all probes were placed in radiologically intact brain tissue (Figure 1H). 

From this point, it became increasingly difficult to ensure adequate PbtO2, and the L/P-ratio remained around 40. CMD-glutamate, which had until then been completely stable around 3–5 µmol/L, increased significantly after the increase in L/P-ratio, and remained high throughout the rest of the measurements (Figure 2D). The reason for the steep increase in glutamate is not known, and beyond being a marker for cellular stress, there is still insufficient evidence on how to interpret the parameter and to use it clinically [9]. Ideally, an EEG could have revealed whether the patient suffered from an epileptic seizure, but this was not available at the time, due to the acute development. We find however, this less likely, as the patient otherwise did not show any tendency to seizures during the entire disease course.

In the end only minor clinical improvement was achieved and even after extensive neurological rehabilitation the patient remains in a vegetative state.

This case illustrates how a gradual decrease in CPP within accepted therapeutic intervals correlated to a marked increase in cerebral anaerobic metabolism, and a shift from mitochondrial dysfunction to an ischemic pattern, followed by clinical worsening.

It exemplifies the clinical relevance of assessing cerebral metabolism through microdialysis, beyond simply using the absolute value of the L/P-ratio as a vague indicator of ischemia or impending vasospasms, and shows how mitochondrial dysfunction as a concept is relevant in the day-to-day care of patients suffering from severe acute brain damage, and how it most likely exists on a continuum with cerebral ischemia [7].

Furthermore, the case demonstrates the difficulty in regulating CPP in aSAH, due to the heterogenic patient population, and the highly dynamic pathophysiology of aSAH, where the optimal perfusion pressure changes over time as the disease progresses. To address this, multimodal neuromonitoring modalities, such as CMD and PbtO2, may reveal important and additional data on the cerebral pathophysiology that are otherwise difficult to assess [7,10].

Future guidelines on CPP regulation in aSAH patients may accordingly benefit by incorporating multimodal neuromonitoring, to provide a dynamic assessment of therapeutic interventions and clinical status, tailored to the individual patient.

CMD = Cerebral microdialysis probe, L = Licox PbtO2 probe, ICP = Camino ICP Probe, EVD = External ventricular drain.

## Figures and Tables

**Figure 1 metabolites-10-00341-f001:**
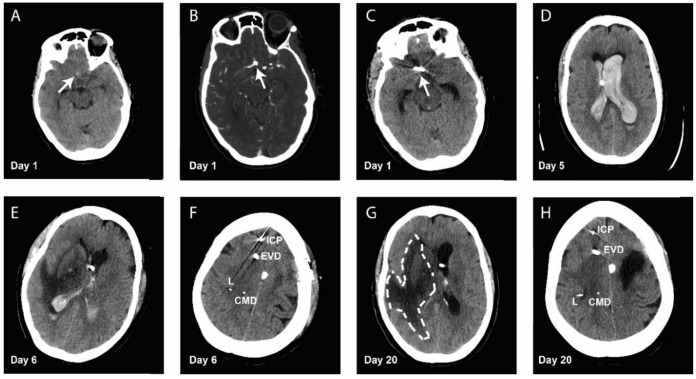
(**A**) CT-scan at admission showing an interhemispheric Fischer grade 1 aSAH bleeding (arrow). (**B**) CT-angiography at admission, showing the aneurism located at the right anterior communicating artery (arrow). (**C**) CT-scan showing a well-placed clip at the aneurism site after the initial operation (arrow). (**D**) CT-scan showing the rebleeding at day 5 after admission. (**E**) Postoperative CT-scan after coiling and evacuation of the intraventricular haematoma. (**F**) Postoperative CT-scan showing probe placement in radiologically intact brain tissue. (**G**) CT-scan at day 20 after admission, showing the growing hypodense area (dashed line). (**H**) CT-scan at day 20 after admission, showing probe placement in radiologically intact brain tissue.

**Figure 2 metabolites-10-00341-f002:**
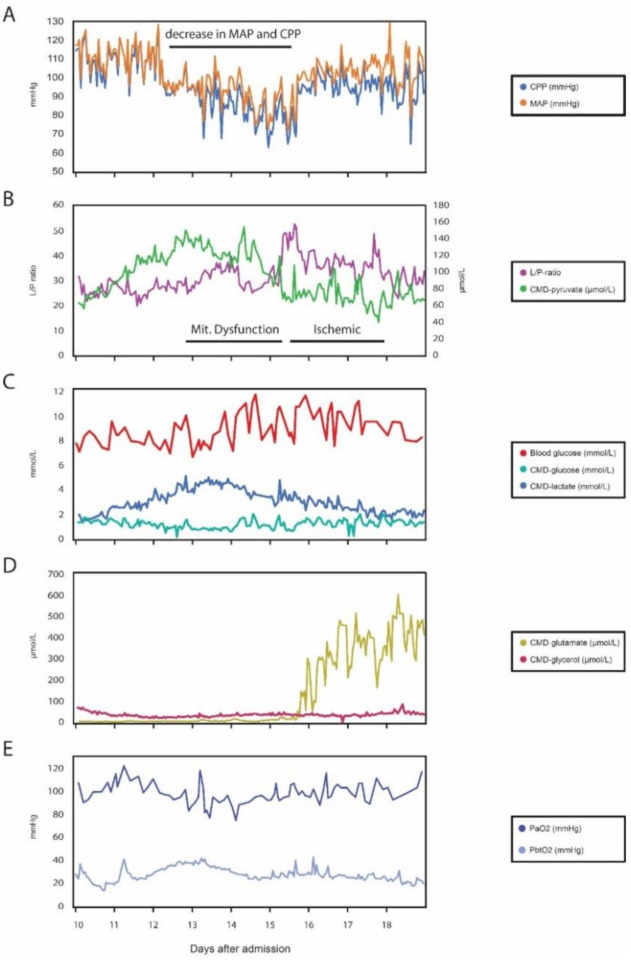
(**A**) Cerebral perfusion pressure (CPP) and mean arterial blood pressure (MAP) were gradually lowered from day 12 after admission. Intracranial pressure (ICP) is the difference between the two curves; it is stable and below 10 mmHg. (**B**) Lactate-pyruvate (L/P)-ratio is seen greatly increasing at day 15 after admission, with a max value of 52.8. Before this a pattern of mitochondrial dysfunction is present (increased L/P-ratio, despite normal or elevated cerebral microdialysis (CMD)-pyruvate). An ischemic pattern is present hereafter (increased L/P-ratio, combined with a decrease in CMD-pyruvate). The difference in both CMD-pyruvate and L/P-ratio was statistically significant (*p* < 0.05), when compared in regard to the time periods before and after the steep increase in L/P-ratio (+/− 2.5 days). (**C**) Blood glucose, CMD-glucose and CMD-lactate. A decrease in CMD-glucose (min. value 0.67 mmol/L) is seen at the same point as the steep increase in L/P-ratio, despite intact blood glucose (7.8 mmol/L). After this time point, blood glucose increased, despite insulin infusion, and became increasingly difficult to stabilize. CMD-glucose increased slightly compared to before the steep L/P-ratio increase. It should be noted that CMD-glucose is dependent on blood glucose. Consequently, high blood glucose can mask low CMD-glucose values. CMD-lactate decreases slightly after day 15, but relatively less than pyruvate, resulting in an increase in L/P-ratio. All changes were statistically significant. (**D**) CMD-glutamate and CMD-glycerol. CMD-glutamate increased sharply after the change from mitochondrial dysfunction to an ischemic pattern at day 15 after admission, possibly indicating cerebral excitotoxicity. CMD-glycerol is stable for the entire period. (**E**) Arterial partial pressure of oxygen (PaO2) and cerebral oxygen tension (PbtO2). From day 15 after admission the PbtO2 decreased, despite a parallel increase in PaO2 and CPP. The difference in PbtO2 was statistically significant. The time resolution for the CMD samples is 1 sample pr. hour.

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
