# Peer review of "Cerebral Microdialysis in Aneurismal Subarachnoid Hemorrhage Patient Reveals a Detrimental Shift in Brain Energy Metabolism, Despite Normal Perfusion Pressure"

_metabolites, 2020, doi:10.3390/metabo10090341_

Round 1

Reviewer 1 Report

Change please abbreviation for Anterior Communicating aneurysm is not common. Use for example ACoA or AcomA. It is not novel the relationship between CCP and mitochondrial disfunction in multimodal monitorization. I think there is a lack of originatity in the case report.

Reviewer 2 Report

In this case study authors depict the challenges often experienced by neurosurgeons and critical care physicians when dealing with aSAH patients. Authors demonstrate their commitment adhering to cutting edge neuromonitoring policies and seek to provide evidence to support the use of invasive monitoring for critical patients. This effort is commendable and merits recognition. However, there are some issues that deserve further explanations and thorough revision by the signing authors.

  • The surgical management of the case might be questionable. A pannel  showing the initial and following scans might be beneficial for the understanding of the actual complications leading to such a poor neurological outcome. Indeed, it should be explained why an endovascular technique was preferred than repositioning the clip and which kind of postoperative control was conducted to neglect a remaining neck or permeable aneurysm.
  • Open surgical evacuation of intraventricular hematomas is very controversial. An EVD and an intratecal fibrinolitc agent after the exclusion of the origin of the bleeding would be the current gold standard.
  • Line 45. Whether the probes were placed in the non-dominant side or ipsilateral to the lesion. The sentence is ambiguous. If the lesion was right, then simply explain how was defined the side of the lesion in an ACom Aneurysm: amount of blood, dominant A1 segment, etc.
  • It should be indicated if the hypodense area was close enough to the probes to be detected. In addition it should be ruled out the relationship between the hypodense area and the surgical/endovascular interventions.
  • In the set of such an increase in glutamate it should be explained that an epileptic status was ruled out. 
  • L/P ratio changes are often a continuum from ischemia to metabolic/mitochondrial impairment. In my opinion such a consideration is worth mentioning.

Overall the paper is well written, worth reading and nicely illustrates events impacting the final outcome of aSAH patients. 

Reviewer 3 Report

The manuscript by Nielsen et al describes a microdialysis study in the brain of a single aneurismal subarachnoid hemorrhage patient. Changes in lactate/pyruvate ratio, pyruvate, glucose, glutamate and glycerol dialysate levels (probe placed ipsilateral to the lesion, watershed area between frontal and middle cerebral artery) are shown up to 13 days after rebleeding.

Cerebral perfusion pressure (CPP), mean arterial blood pressure (MAP), blood glucose levels, arterial partial pressure of oxygen (PaO2) and cerebral oxygen tension (PbtO2) were also monitored and data are shown.

The used methods to measure brain glucose metabolism are standard in neurointensive care units, employing an enzymatic-fluorometric assay (Iscus flex, Mdialysis). It is not clear what the time resolution is of the clinical microdialysis set-up, but the variation in the figure suggests that this may be one hour or longer.

The paper may be of interest to neurosurgeants, but it is of little interest to readers of Metabolites. The employed methods are not new, only a single case is described and the data and interpretation are not very convincing.

For instance fig B:  Before day 10 the metabolic pattern is labeled “mitochondrial dysfunction” and after day 10 “ischemic”. This is based on a higher L/P ratio and lower pyruvate level after day 10, but according to the asterisk in the figure only a few samples between day 10 and 11 have a significantly elevated L/P ratio (what kind of statistic test was used?). Are changes in pyruvate levels before and after day 10 significantly different? If the period after day 10 is considered “ischemic”, I would expect no or very low glucose (fig. C) and oxygen (fig. E) levels, which is not apparent from the figures (the figures show almost no change). Glutamate levels rise (fig D), which may indicate compromised energy metabolism, but since glycerol levels do not change, there is probably no (excitotoxic) cell death. All together, I find the data not sufficiently discussed.

Minor points: Avoid abbreviations in abstract. Explain abbreviations at first use in the text – not all readers are familiar with ICU-jargon.

Round 2

Reviewer 1 Report

Authors improved the paper. It ‘s not a novel issue,  but the authors explained correctly the interest of the paper. 

Reviewer 2 Report

Recommendations and suggestions have been properly adressed.

I would like to congratulate the authors for their contribution and encourage them to keep working on this line.

Reviewer 3 Report

Comments and suggestions for improvement of metabolites-848563-v2

Title:

  • A person can “recognize” a shift, not a technique (microdialysis). Use “reveals” or “shows”
  • aSAH should be spelled out.
  • Three times “cerebral” is too much.
  • “Accepted” by whom? Use “normal”.

Suggested new title:

Cerebral microdialysis in aneurismal subarachnoid hemorrhage patient reveals a detrimental shift in brain energy metabolism despite normal perfusion pressure.

Abstract:

No abbreviations in the abstract, thus:

ICU should be spelled out

aSAH should be spelled out

CPP should be spelled out

I am confused by the conclusion

“The case highlights the importance and difficulty in regulating CPP amomg aSAH patients”

I think this case shows that dispite regulating CPP (which is a common intervention in neuro-ICU), brain metabolism and clinical outcome worsened. Thus, in this case

 regulating CPP was not enough.

 The authors should rewrite the conclusion in the abstract.

Main text:

Arteria communicans anterior should be written in English, thus rewrite like this:

broad based aneurism of the anterior communicating artery

GCS should be spelled out. GCS were? Is GCS plural? Should this be: Global consciousness scale rating was 15, which is in the normal range?

The main text with description of time course and clinical situation of the patient – days after rebleeding - , clinical interventions (manipulating MAP and CPP) and various monitoring techniques (brain oxygen, ICP, microdialysis) is difficult to follow (lines 52 -78).

It would be helpful to discuss to the different interventions and subsequent alterations in physical and biochemical parameters by referring to the relevant time intervals in figure 1A, B, C, D or E. Place the figures A-E  in the same order as the different parameters are discussed in the main text.

The authors responded to my first comments that “Applying a statistical test would be meaningless and also misleading in this case report with only one patient and with data that should be assessed dynamically.”

I do not agree and suggest that the authors perform non-parametric t-tests to assess whether the different parameters (L/P ratio, pyruvate, lactate, glucose, oxygen) in the period 5-10 days, labelled “mitochondrial dysfunction” and the period 10-14 days, labelled “ischemic”, are significantly different.  I recommend the authors to do this for all of the measured variables, so they can discuss which parameters are most useful in monitoring a decline in cerebral metabolism.

The authors should also briefly describe and discuss the concept of using the L/P ratio in combination with elevated pyruvate levels as an index of mitochondrial dysfunction. This is hardly discussed.

The authors should consider showing the lactate data (mM levels) in figure 1B, since these may be more important to reveal hypoxia/ischemia than the (less reliable) changes of pyruvate  (between 60 and 140 µM).

As mentioned in my first comment to the ms, microdialysate glucose and PbtO2 changes do not reveal an ischemic period. This should be discussed.

In figure 2 it is shown that the CMD probe is placed in intact brain tissue. Thus, there is no ischemia in this area of the brain? Please discuss.

Figure 1: Legend should include statistical test results of periods before and after day 10.

Figure 2: Figures (CT-scans) should be labelled A, B, C etc. and a legend for the whole figure should be given. Separate legends for each CT-scan should be removed. Use arrows or asterisk to label the relevant structures on the CT-scans – some are difficult to find without help. Use only scans with relevant information.

Round 3

Reviewer 3 Report

All comments and concerns on previous versions of the manuscript have been addressed and the manuscript has been significantly improved. I think the present version is suitable for publication in Metabolites.